# Molecular Dynamics Simulations for Effects of Fluoropolymer Binder Content in CL-20/TNT Based Polymer-Bonded Explosives

**DOI:** 10.3390/molecules26164876

**Published:** 2021-08-12

**Authors:** Shenshen Li, Jijun Xiao

**Affiliations:** Molecules and Materials Computation Institute, School of Chemistry and Chemical Engineering, Nanjing University of Science and Technology, Nanjing 210094, China; lishenshen91@163.com

**Keywords:** CL-20/TNT co-crystal, fluoropolymer binder, interactions, mechanical properties, polymer-bonded explosives (PBXs), molecular dynamics (MD) simulation

## Abstract

In order to better understand the role of binder content, molecular dynamics (MD) simulations were performed to study the interfacial interactions, sensitivity and mechanical properties of 2,4,6,8,10,12-hexanitro-2,4,6,8,10,12-hexaazaisowurtzitane/2,4,6-trinitrotoluene (CL-20/TNT) based polymer-bonded explosives (PBXs) with fluorine rubber F_2311_. The binding energy between CL-20/TNT co-crystal (1 0 0) surface and F_2311_, pair correlation function, the maximum bond length of the N–NO_2_ trigger bond, and the mechanical properties of the PBXs were reported. From the calculated binding energy, it was found that binding energy increases with increasing F_2311_ content. Additionally, according to the results of pair correlation function, it turns out that H–O hydrogen bonds and H–F hydrogen bonds exist between F_2311_ molecules and the molecules in CL-20/TNT. The length of trigger bond in CL-20/TNT were adopted as theoretical criterion of sensitivity. The maximum bond length of the N–NO_2_ trigger bond decreased very significantly when the F_2311_ content increased from 0 to 9.2%. This indicated increasing F_2311_ content can reduce sensitivity and improve thermal stability. However, the maximum bond length of the N–NO_2_ trigger bond remained essentially unchanged when the F_2311_ content was further increased. Additionally, the calculated mechanical data indicated that with the increase in F_2311_ content, the rigidity of CL-20/TNT based PBXs was decrease, the toughness was improved.

## 1. Introduction

High-performance and insensitive explosives are always the target of researchers in the field of energetic materials [1,2,3,4]. However, high performance and safety are somewhat mutually exclusive for current single compound explosives, which seriously limits their development and applications [5,6]. Among well-known commercially available single-compound explosives, 2,4,6,8,10,12-hexanitro-2,4,6,8,10,12-hexaazaisowurtzitane (CL-20) is the most famous high energy density compound [7], which features high density and high detonation velocity but fails to meet the important safety requirements due to its high sensitivity [8]. Contrasting with CL-20, 2,4,6-trinitrotoluene (TNT) has different features in many ways [9,10,11,12], including low oxygen balance, modest detonation velocity, economical production costs, and low impact sensitivity, but energy density is relatively low. Fortunately, the method of producing co-crystals offers a practical solution to improve certain properties of energetic materials such as oxygen balance, sensitivity, detonation velocity, and safety [13,14,15]. Bolton [16] prepared co-crystal of CL-20/TNT in a 1:1 molar ratio, which exhibits lower sensitivity than CL-20, and has closed detonation performance and oxygen balance to CL-20.

The combination of explosives with polymeric binders to form polymer-bonded explosives (PBXs) was an important advancement in high-explosives science, offering improved safety and reliability, while maintaining performance [17,18,19]. Based on those advantages, PBXs are widely applied in many defense and economic scopes. A lot of experimental research [8,20] and theoretical studies [21,22,23] on energetic composite materials including PBXs are drawing more and more attention in recent decades. Not only can PBXs reduce the impact and friction sensitivity of explosives in PBXs [24,25], but they also have good physical and mechanical properties of polymers [26,27] and hence can be produced and used safely and conveniently.

Due to a balance of chemical and mechanical properties with processability when mixed with explosives, fluoropolymers emerged as commonly used binder [28]. To estimate the effect of the polymer binders on the co-crystal-based PBXs, we select fluoropolymer F_2311_ as the polymeric binder in CL-20/TNT based PBXs, which is a random copolymer made up of vinylidenedifluoride (VDF) and chlorotrifluoroethylene (CTFE) with the molar ratio of 1:1, showed in Figure 1a.

The goal of this study is to explore intermolecular interactions and mechanical properties of the CL-20/TNT-based PBXs in different F_2311_ contents using molecular dynamics (MD) simulation. This paper is arranged as follows. At first, several CL-20/TNT based PBXs in different F_2311_ content were constructed. Then, we performed MD simulations to study interactions between CL-20/TNT and F_2311_ and the mechanical properties of PBXs. Pair correlation function (PCF) was used to explore the interface structures between the co-crystal explosive and the polymer binders. Binding energy (*E*_bind_) can provide information about interfacial reaction and thermal stability. More specifically, N-NO_2_ trigger bond lengths of CL-20 were discussed in terms of the relationship with sensitivity. In addition, the mechanical properties such as tensile modulus (*E*), bulk modulus (*K*), shear modulus (*G*), Poisson’s ratio (ν) and Cauchy pressure (*C*_12_-*C*_44_) were also calculated, and both are discussed in this paper.

## 2. Models and Computational Methods

All simulations were conducted utilizing the condensed-phase optimized molecular potentials for atomistic simulation studies (COMPASS) force field [29], which is suitable for simulations of nitro-compound explosives and their PBXs, and is suitable for calculating of interfacial interactions between different components in PBXs [30,31,32]. The primitive CL-20/TNT co-crystal cell [16] derived from X-ray diffraction contains 8 CL-20 molecules and 8 TNT molecules. Based on the co-crystal cell, the primary cell of CL-20/TNT corresponding to 6 (3 × 2 × 1) unit cells were built. We built four F_2311_ molecular chains, which contain 10, 36, 62 and 88 constitutional repeating units, respectively. Additionally, the corresponding amorphous F_2311_ cells were obtained by using the high-low pressure dynamics simulation method [33]. The procedure of building PBXs models were as follows. For the crystalline surfaces (1 0 0) of the CL-20/TNT cocrystal (3 × 2 × 1) unit cells, the *c* lattice length was 43.40 Å, and the corresponding interface, *a* × *b*, was 26.74 Å × 24.70 Å. Then, the corresponding interface, *a* × *b*, of the F_2311_ cells was changed to 26.74 Å × 24.70 Å; the size *a* × *b* was kept unchanged; *c* lattice lengths of fluoropolymer box step-by-step were resized until the F_2311_ theoretical density was reached; each step of the adjustment needs the molecular dynamics simulation running to equilibrium state. Then, PBXs models were made by merging F_2311_ fluoropolymer on the (1 0 0) crystalline surface of CL-20/TNT. Therefore, there are four PBXs models shown in Figure 2. The weight percentages of the F_2311_ in the PBXs models are 2.5%, 9.2%, 14.9% and 19.9%, respectively.

All the models were allowed to evolve dynamically in isothermal-isobaric (NPT) ensemble at 300K and atmospheric pressure, in which the temperature was maintained through the Andersen stochastic collision method [34] and the pressure was controlled via the Parrinello–Rahman [35] scheme with all cell parameters fully relaxed at atmospheric pressure. The van der Waals (vdW) interactions were truncated at 9.5 Å with long range tail correction, and the electrostatic interactions were calculated via the standard Ewald summation. The equations of motion were integrated with a step of 1 fs. Equilibration run was performed for 5 ns, which is referred to the equilibrium at the new state by running for a period for the simulating model and should be extended at least until the instantaneous values of the potential energy and temperature, etc., for the simulating model have ceased to show a systematic drift and have started to oscillate about steady mean values. After equilibration run, production runs of 1 ns were performed, during which data were collected with 10 fs sampling interval for analysis. These computations were all carried out using software Material Studio from Biovia Inc.

## 3. Results and Discussions

### 3.1. Binding Energy

Binding energy (*E*_bind_) is defined as the negative value of the intermolecular interaction energy (*E*_inter_). The intermolecular interaction energy between different components can calculate by subtracting individual component energy in the system from the total energy of the whole system [30]. As mentioned above, *E*_bind_ between CL-20/TNT and F_2311_ can be evaluated as *E*_bind_ = −*E*_inter_ =−(*E*_total_ − *E*_CL-20/TNT_ − *E*_F2311_), where *E*_total_ is the PBXs total energy, *E*_CL-20/TNT_ and *E*_F2311_ are the energy of CL-20/TNT and F_2311_, respectively. Binding energy stands for the level of interaction between two components. In this paper, it can reflect the thermal stability of energetic systems and can find out the influence on PBXs with different F_2311_ contents. *E*_total_, *E*_CL-20/TNT_, *E*_F2311_, *E*_bind_ and *E*_bind_′ are tabulated in Table 1, where *E*_bind_′ is the *E*_bind_ of unit quantity of F_2311_ binders.

Based on the theoretical results shown in Table 1, we can find that the value of *E*_bind_ increases with increase in F_2311_ contents in PBXs, which means that increasing the F_2311_ content in CL-20/TNT-based PBXs can enhance the interaction between F_2311_ and CL-20/TNT and the thermodynamic stability of the PBXs. However, it was noticed that the trend of *E*_bind_’ decreases with F_2311_ contents. Herein, we can find a reasonable explanation from the illustration in Figure 2, where it can be seen that the quantity of uncontacted F_2311_ with CL-20/TNT increased more with increasing of F_2311_ contents than the quantity of contacted F_2311_.

### 3.2. Pair Correction Function

The interface structure between CL-20/TNT and F_3211_ molecular chain was explored by pair correlation function (PCF). PCF gives a measure of probability density *g*(r) of finding an atom at some distance and thus provides insight into a material structure through revealed local spatial ordering. The PCF curves for different atom pairs in CL-20/TNT and F_3211_ were plotted in Figure 3. The oxygen atoms in CL-20/TNT, nitrogen atom in CL-20/TNT and fluorine atoms in F_2311_ were labeled as O, N and F, respectively, while hydrogen atoms in F_2311_ and CL-20/TNT were labeled as H1 and H2, respectively.

Generally the interaction distance range for hydrogen bond is 2.0–3.1 Å, and the distance range for stronger vdW and electrostatic interactions is 3.1–5.0 Å. When the distance between two atoms is farther than 5.0 Å, the vdW interaction is quite weak. From Figure 3a,b, it was found that in hydrogen bond range, the PCF curves all give comparatively high peaks, indicating that the hydrogen bonds exist in H_2_···F pairs and H_1_···O pairs of F_2311_ molecules and CL-20/TNT. Additionally, the peaks vary in intensity with F_2311_ content. From Figure 3a, we can find that the *g*(r) value of low F_2311_ content is mostly larger than high F_2311_ content. Similarly, the trend of *g*(r) value in Figure 3b is almost the same, but largest value for hydrogen bonds is 9.2% F_2311_ content. Figure 4 gives a visual representation of hydrogen bonding interactions of H_1_···O and of H_2_···F for 2.5% F_2311_ content, for example. For H_1_···N in Figure 3c, the curve has a comparatively high peak only in the vdW interaction range, implying only vdW and electrostatic interactions exist between H_1_···N.

### 3.3. N-NO_2_ Trigger Bond Length

For energetic compounds, there exists a criterion to theoretically judge the relative sensitivity [22]. According to Principle of Smallest Bond Order (PBSO) based on quantum chemical calculation, for a series of energetic compounds with smaller bond order of trigger bond in molecular means the compound is more sensitive [36,37]. This principle has been used extensively in the prediction of impact sensitivity for various types of energetic compounds. Classical MD simulation does not provide electronic structure, and cannot give the bond order data. However, MD simulation can provide statistical distribution of bond length instead of the bond order data. Usually, chemical bond length can be characterized by the bond order and bond length in molecular. Thus it is suitable to evaluate sensitivity based on the molecular structure parameter, bond length, obtained through MD simulation.

We all know that the N-NO_2_ bond is a trigger bond of nitramine explosives. Additionally, it is well know that CL-20 is more sensitive than TNT, and the CL-20 component in CL-20/TNT is prior to decompose in detonation. Therefore, in this work, the N-NO_2_ bond in CL-20 is chosen as the trigger bond. Table 2 presents the results of trigger bond (N-NO_2_) lengths of CL-20 component in PBXs at different F_2311_ contents. When the bond length is longer, the bond is fractured more easily, which makes CL-20 molecules more active and decompose more easily. It can be found from Table 2 that the average bond lengths *L*_ave_ of all the models are almost unchanged, but the maximum bond length (*L*_max_) of trigger bond has more obvious change with F_2311_ content increasing, which means that *L*_max_ of the trigger bond is more tightly correlated to the initial bond fractured in PBXs’ detonation. As the content increases, the maximum bond length decreases gradually. When the content of F_2311_ rose to 9.2%, the *L*_max_ decreased from 1.5972 Å to 1.5817 Å. After that, the *L*_max_ changed slightly, basically maintaining the level of 9.2%. The change is consistent with the fact that energetic material becomes less sensitive as F_2311_ content increases. Hence, using *L*_max_ to measure the sensitivity of energetic is efficient. Additionally, we can find that when the content of F_2311_ reaches 9.2%, continuous increase of the F_2311_ content has less effect on the sensibility.

### 3.4. Mechanical Properties

Mechanical properties are the most important properties of energy materials, because they are related to the preparation, storage, transportation and usage of materials. Elastic modulus is the index of a material’s stiffness, and it is also the measurement of a material’s resistance to elastic deformation. The plasticity and fracture properties can be related to the elastic modulus. Higher shear modulus means higher stiffness and shear strength, reflecting resistance to shearing strain [23,38,39]. Higher bulk modulus means higher rupture strength, that is, the greater the value of *K* is, the more energy will be required for a material to rupture [31,40,41]. Cauchy pressure (*C*_12_-*C*_44_) can reflect the brittle/ductile behavior of a material. The high value of Cauchy pressure (*C*_12_-*C*_44_) is related to ductility, and the low value is related to brittleness. When a material is compressed in one direction, it usually tends to expand in the other two directions perpendicular to the direction of compression. Poisson’s ratio is a measure of this effect.

Based on the fluctuation analysis of the production trajectories and Reuss average [42] of the co-crystal model (without F_2311_ binder) and its corresponding PBXs models, the calculated tensile modulus (*E*), bulk modulus (*K*), shear modulus (*G*), Poisson’s ratio (*ν*), Cauchy pressure (*C*_12_-*C*_44_) are listed Table 3. As we can see from Table 3, *E*, *K* and *G* reduced gradually with the increase of F_2311_ content, indicating that the stiffness of the PBXs decreases. Additionally, it can be found that the polymer addition has less effect on Poisson’s ratio. The Cauchy pressure (*C*_12_-*C*_44_) values increase with F_2311_ content increasing, which can be deduced that the ductility of the Cl-20/TNT based PBXs increases with increasing F_2311_ content. Generally, less stiffness and better ductility for PBXs compared to CL-20/TNT means it is easier to deplete and disperse partially the external stimulus energy imposed on them during loading and transportation, which can reduce possibility of hot spots, and thus has lower sensitivity.

## 4. Conclusions

In this study, we performed a NPT-MD simulations of CL-20/TNT-based PBXs with F_2311_ as polymeric binders. The simulations involved binding energy calculation and PCF analysis for the thermal stability evaluation and the interfacial structure exploration between the co-crystal and F_2311_, the maximum bond length of the N–NO_2_ trigger bond and mechanical property computation. These studies are in favor of theoretical research and practical applications of the co-crystal.

From the calculated binding energies, *E*_bind_ between F_2311_ and Cl-20/TNT increases with the increasing content of F_2311_, but *E*_bind_ of unit quantity of F_2311_ binders reduces due to the specific surface area of F_2311_ contacted CL-20/TNT. Therefore, the interaction between the binder and the explosive can be improved by increasing the contact area of binder and explosive. PCF analysis of atom pairs in the interfacial structure has indicated that hydrogen bond exists between CL-20/TNT and F_2311_. Additionally, the hydrogen bond mainly came from H1···O and of H2···F. By analyzing the maximum bond length of N-NO_2_ trigger bond, it can be found that increasing F_2311_ content can decrease the sensitivity of CL-20/TNT based PBXs, but a continuous increase in the F_2311_ content has less effect on the sensibility when the content of F_2311_ reaches 9.2%. The mechanical properties shows that as the F_2311_ content increases, the moduli of CL-20/TNT-based PBXs decrease, but the ductility was improved. To sum up, the small amount of polymer binders F_2311_ coating with the CL-20/TNT co-crystal makes the PBXs more insensitive and give them better mechanical properties.

## Figures and Tables

**Figure 1 molecules-26-04876-f001:**
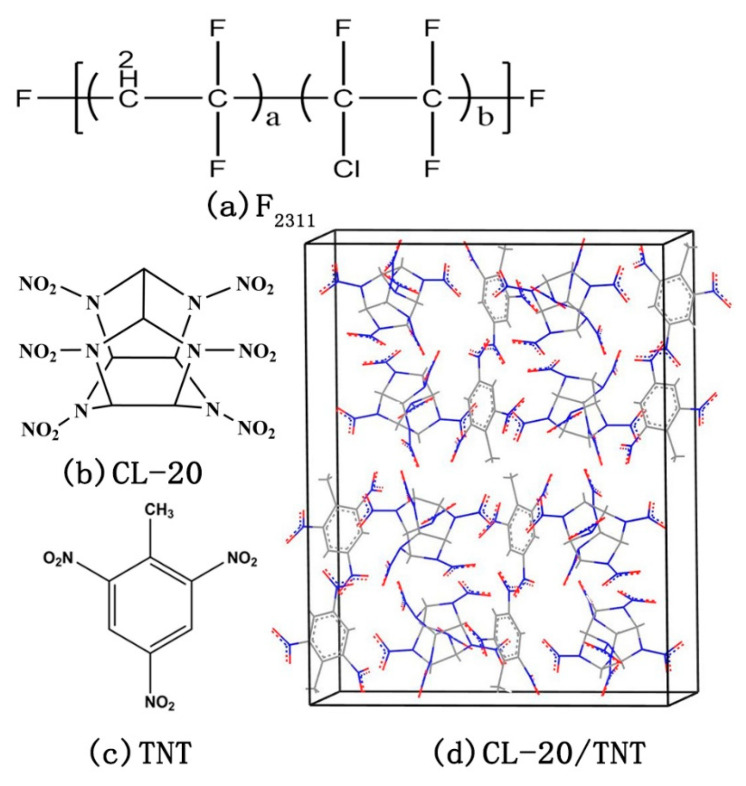
Molecular structures of (**a**) F2311, (**b**) CL-20, and (**c**) TNT. (**d**) The primitive cell of the CL-20/TNT co-crystal.

**Figure 2 molecules-26-04876-f002:**
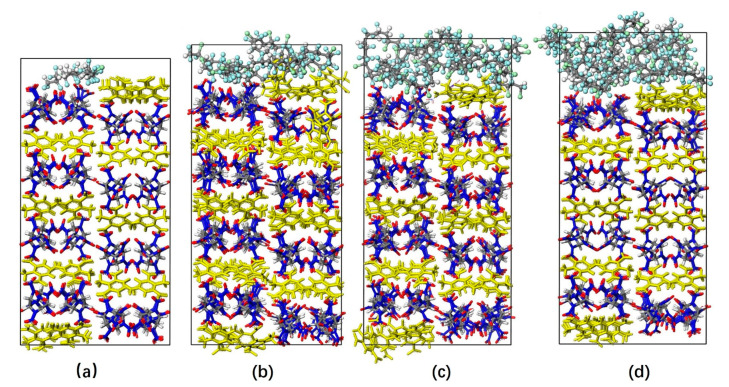
PBXs models with F_2311_ content (**a**) 2.5%, (**b**) 9.2%, (**c**) 14.9%, (**d**) 19.9% (CL-20 in stick model, TNT in yellow and F_2311_ in Ball model).

**Figure 3 molecules-26-04876-f003:**
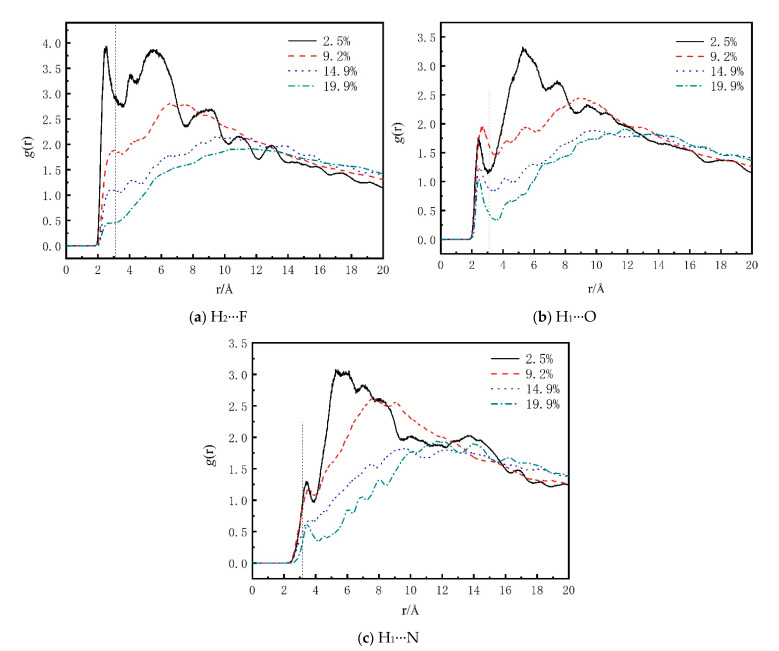
PCF for atom pairs in CL-20/TNT-based PBXs, (**a**) H_2_···F, (**b**) H_1_···O, (**c**) H_1_···N.

**Figure 4 molecules-26-04876-f004:**
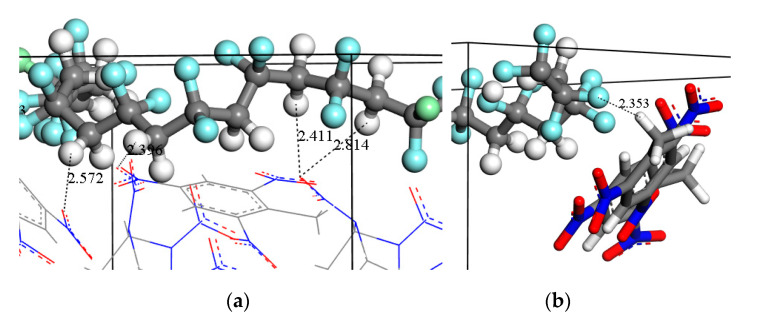
Illustration of hydrogen bonds of (**a**) O atoms in the co-crystal (red line model) with H atoms in F_2311_ (white ball) and (**b**) H atoms in the co-crystal (white stick) with F atoms in F_2311_ (pale blue ball).

**Table 1 molecules-26-04876-t001:** *E*_bind_, *E*_total_, *E*_F2311_, *E*_CL-20/TNT_ and *E*_bind_′ of PBXs in different F_2311_ contents *^a^*.

F_2311_ Contents	*E* _bind_	*E* _total_	*E* _F2311_	*E* _CL-20/TNT_	*E*_bind_′
2.5%	90.64	−1309.09	−325.22	−1267.50	36.25
	(7.13)	(30.64)	(9.07)	(31.27)	(1.59)
9.2%	250.95	−1341.98	−643.64	−1252.52	27.27
	(4.47)	(30.82)	(9.16)	(28.65)	(1.26)
14.9%	281.69	−1393.27	−1137.96	−1251.30	18.90
	(4.59)	(29.36)	(7.25)	(23.24)	(1.43)
19.9%	356.72	−1477.70	−923.94	−1256.59	17.93
	(6.75)	(29.99)	(7.82)	(31.93)	(1.22)

*^a^* Unit: kcal·mol^−^^1^; The corresponding deviations are listed in parenthesis.

**Table 2 molecules-26-04876-t002:** The trigger bond (N-NO_2_) lengths (Å) of CL-20 at different F_2311_ contents *^a^*.

Content	0.0%	2.5%	9.2%	14.9%	19.9%
*L* _ave_	1.3927	1.3935	1.3936	1.3932	1.3934
	(0.028) *^a^*	(0.028)	(0.031)	(0.030)	(0.030)
*L* _max_	1.5972	1.5843	1.5817	1.5816	1.5817

*^a^* The corresponding deviations for *L*_ave_ are listed in parenthesis.

**Table 3 molecules-26-04876-t003:** Tensile modulus (*E*), bulk modulus (*K*), shear modulus (*G*), Poisson’s ratio (*ν*) and Cauchy pressure (*C*_12_-*C*_44_) for the PBXs models with different F_2311_ contents *^a^*.

Content	0.0%	2.5%	9.2%	14.9%	19.9%
*E*	6.22(0.03)	5.38(0.05)	4.32(0.06)	4.11(0.03)	3.65(0.02)
*K*	7.49(0.04)	6.41(0.10)	5.22(0.12)	5.02(0.07)	4.54(0.04)
*G*	1.88(0.02)	1.54(0.03)	1.34(0.07)	1.36(0.02)	1.33(0.02)
*ν*	0.38(0.02)	0.39(0.00)	0.38(0.00)	0.38(0.00)	0.37(0.00)
*C*_12_-*C*_44_	1.79(0.11)	2.81(0.11)	2.78(0.16)	3.07(0.13)	3.42(0.11)

*^a^* The corresponding deviations are listed in parenthesis. The units for *E, K* and *G* are GPa.

## Data Availability

All data are contained within the article.

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
