# Peer review of "Molecular Dynamics Simulations for Effects of Fluoropolymer Binder Content in CL-20/TNT Based Polymer-Bonded Explosives"

_molecules, 2021, doi:10.3390/molecules26164876_

Round 1
Reviewer 1 Report
This manuscript reports molecular dynamics (MD) simulations to investigate the interfacial interactions, sensitivity, and mechanical properties of CL-20/TNT with fluorine rubber F_2311. The manuscript is well written and deserves publication in the Molecules after minor corrections.
- Figure 1. (d). Increase the figure's size to visualize better the primitive cell of the CL-20/TNT co-crystal.
- If possible, improve the quality of Figure 2 (a)-(d). Although Figure 4 shows the H-bonds between O atoms of co-crystal and H-atoms of F_2311 and the interaction of H-atoms in the co-cristal and F-atoms of F_2311, Figure 2 (a)-(d) not shown clearly these weak bonds at different F2311 concentrations.
- Tables 1-3. Change the word "brackets" to "parenthesis".
Reviewer 2 Report
This paper discusses the molecular dynamics modeling of CL-20/TNT systems with varying levels of fluoropolymer content. The trigger bond lengths and mechanical properties are predicted. The results indicate that the fluoropolymer content has a significant effect on these properties. There are serious issues with this paper:
- In general, the manuscript is difficult to read due to poor English grammar and word choice
- The authors provide no explanation of why they are modeling a layered structure, or how they choose the relative content of the fluoropolymer in the MD simulation box.
- The length of the equilibration runs is not provided
- The authors make an absurd statement: “Higher shear modulus means higher hardness and yield strength”. Not only is this not true, but it indicates a serious lack of understanding of solid mechanics on the part of the authors
- Another absurd statement: “Higher bulk modulus means higher fracture strength”. Again, this statement is completely untrue and indicates a fundamental lack of understanding of solid mechanics
Round 2
Reviewer 2 Report
The authors have satisfactorily addressed my comments